# Emotional and Physical Symptoms Following Intimate Partner Violence Victimization in the United States: Implications for Law and Public Health Policy

**DOI:** 10.3390/ijerph22121829

**Published:** 2025-12-05

**Authors:** Gia Elise Barboza-Salerno, Karla Shockley McCarthy, Taylor Harrington, Amy Watson-Grace

**Affiliations:** 1Colleges of Social Work and Public Health, The Ohio State University, Columbus, OH 43210, USA; 2College of Public Health, The Ohio State University, Columbus, OH 43210, USA; harrington.448@buckeyemail.osu.edu; 3Ohio Colleges of Medicine Government Resource Center, Columbus, OH 43210, USA; karla.shockleymccarthy@osumc.edu; 4School of Health and Rehabilitation Sciences, The Ohio State University, Columbus, OH 43210, USA; amy.watson-grace@osumc.edu

**Keywords:** intimate partner violence (IPV), traumatic brain injury (TBI), National Crime Victimization Survey (NCVS), emotional symptoms, physical symptoms, race/ethnicity

## Abstract

**Highlights:**

**Public health relevance—How does this work relate to a public health issue?**

**Public health significance—Why is this work of significance to public health?**

**Public health implications—What are the key implications or messages for practitioners, policy makers and/or researchers in public health?**

**Abstract:**

Intimate partner violence (IPV) is a public health issue that produces significant psychological and physiological consequences. This exploratory descriptive study examines whether sustaining a serious injury increases the likelihood that IPV survivors experience emotional and physical symptoms. We analyzed nonfatal violence by an intimate partner reported in the U.S. National Crime Victimization Survey (2009–2023) and applied survey-adjusted logistic regression models. We assessed two dichotomous outcomes: (1) whether respondents reported at least one emotional symptom (e.g., vulnerable, violated, distrustful, or unsafe) and (2) whether they reported at least one physical symptom (e.g., headaches, fatigue, or muscle tension). We define serious injury as broken bones, gunshot wounds, internal injuries, or loss of consciousness. We included demographic characteristics (age, race, sex, and educational attainment) as control variables. The results show that IPV survivors who sustained serious injuries had significantly higher odds of reporting both emotional and physical symptoms than those who did not sustain such injuries. These findings underscore how serious injury compounds the burden of IPV and emphasize the need for comprehensive medical, legal, and psychosocial interventions to address its lasting health impacts.

## 1. Introduction

Intimate partner violence (IPV) is a pervasive public health problem in the United States [1,2,3,4]. IPV affects people of all genders, sexual orientations, and relationship types, and includes threatened, attempted, or completed physical or sexual violence, emotional abuse, and stalking by a spouse, partner, or dating partner [1,5,6]. Although both men and women experience IPV, women bear a disproportionate burden, facing higher rates of victimization along with a much greater risk of injury and death [6,7]. Women are significantly more likely than men to report IPV-related rape, physical assault, or stalking, and they experience more chronic and injurious assaults, with more than 40% reporting injury during their most recent incident compared to fewer than 20% of men [5]. One of the most extreme forms of gender-based violence is pregnancy-associated femicide, where intimate partners murder women during or after pregnancy, accounting for up to 41% of maternal homicides [1,7]. Research further shows that between 6.7% and 17.2% of males experience IPV [8,9]. Nevertheless, men are frequently omitted from analyses, which complicates our understanding of the scope and consequences of IPV.

Abused women face worse overall physical health, lower quality of life, and higher use of medical services compared to women who have not been abused [1,10,11]. Injuries frequently affect the head, face, and neck, but may also include lacerations, broken bones, and internal damage [12]. Beyond immediate harm, IPV produces chronic physical conditions such as headaches, back pain, and musculoskeletal issues [1]. Fatigue, insomnia, nightmares, and restless sleep commonly follow IPV, exacerbating depression and distress [13]. Longer-term consequences of IPV victimization include cardiovascular problems, hypertension [11], and disordered eating habits [6]. In the most severe cases, IPV causes traumatic brain injury (TBI), with symptoms ranging from loss of consciousness and headaches to long-term neurological impairment [12]. Women who experience sexual assault-related IPV face a higher risk of injuries, chronic pain, gastrointestinal and gynecological problems, sexually transmitted infections, and mental health conditions such as depression and PTSD [1,14]. A 2025 systematic review reported consistent associations between IPV and depression in 10 of 11 studies, most of which relied on hospital, shelter, or police samples. While these settings provide valuable insights, they limit generalizability because they exclude survivors who never seek formal services.

Experiencing violence at the hands of an intimate partner constitutes a betrayal of trust and represents a distinct form of emotional injury [15]. This experience aligns with the concept of betrayal trauma [16]. Limited research on betrayal trauma suggests that it creates barriers to forming healthy relationships by fostering fear, shame, and low self-esteem. It also erodes trust and security, producing ongoing feelings of vulnerability, violation, unsafety, and shame [4]. Shame is defined as a negative emotion or painful experience that results in feelings of mistrust, vulnerability, fear of stigmatization, and self-blame [17,18,19,20]. Shame operates not only as an immediate emotional response but also as a longer-term psychosocial consequence of IPV, functioning as a barrier to help-seeking among female survivors [18]. When survivors do seek help, they often encounter individuals or institutions that reinforce rather than reduce this shame. Ethno-racial identity can compound these dynamics, as cultural and social identities shape how shame is experienced and expressed in help-seeking contexts [21].

Despite decades of research on IPV across medicine, psychology, and criminology, important gaps remain. Most studies have focused on general mental and physical health outcomes among survivors, documenting conditions such as depression, PTSD, or overall physical health restrictions. However, these outcomes are rarely linked directly to the IPV incident itself, making it difficult to understand the specific symptoms that emerge in the immediate weeks and months after victimization. A recent review highlights that although IPV affects women’s physical health well beyond acute injuries and mental health consequences, few studies validate these findings, underscoring the need for more research on the long-term effects of IPV, especially across different subtypes [22]. The review further notes that, despite challenges, such research is essential to establish links between IPV and chronic health conditions, thereby furthering the evidence base to establish IPV as a critical health issue.

To our knowledge, no prior study has examined both emotional and physical symptoms following intimate partner violence (IPV) using a nationally representative sample that links symptom persistence directly to the specific incident of partner violence and associated injury, including probable traumatic brain injury (pTBI), while controlling for sociodemographic and contextual factors. The present study addresses this gap by focusing exclusively on IPV, defined by the Bureau of Justice Statistics (BJS) as violence committed by a current or former spouse, boyfriend, or girlfriend, including sexual or physical assault [23,24]. Using this definition, we assess how injury severity, particularly loss of consciousness as a marker of pTBI, affects the likelihood of emotional and physical symptoms such as headaches, fatigue, gastrointestinal issues, anxiety, mistrust, and feelings of being unsafe that persist for at least one month following an IPV incident within the past six months. Specifically, this study has two primary objectives: first, to determine the national prevalence of emotional and physical symptoms following IPV across key sociodemographic groups; and second, to examine whether injury, including probable traumatic brain injury, exacerbates these risks beyond sociodemographic characteristics. By doing so, this study provides the most comprehensive evidence to date on the short- to medium-term emotional and physical consequences of IPV in the United States, with implications for both public health and legal systems.

## 2. Materials and Methods

Data for this study were analyzed from incident-level records from the NCVS public-use files. The NCVS is a nationally representative household survey administered by the BJS [25]. The NCVS conducts interviews with over 90,000 households twice annually, with sampled households remaining in the study for a period of three years. Respondents aged 12 and older provide information on demographics, household composition, and victimizations that occurred in the past six months, along with follow-up questions about offender characteristics, incident context, injuries, police reporting, and consequences of IPV victimization. We limited our analysis to survey years 2009 through 2023, as questions about emotional and physical symptoms after victimization were fully included starting in 2009.

We followed the coding scheme of others [26] to identify incidents of IPV. Specifically, we coded cases in which offenders were reported as spouses, ex-spouses, or boyfriends/girlfriends, or when multiple offenders included a partner. We excluded victimizations committed solely by other family members or strangers. Following the BJS classification, we further limited the analytic sample to violent crimes, defined as sexual violence and assault (including both simple and aggravated assault). This resulted in a weighted, nationally representative sample of IPV incidents during the study period. For descriptive analyses, we distinguished between two operational definitions of IPV used in prior research: (1) a broad definition, which includes all violent victimizations committed by intimate partners (such as robbery or threats in addition to assault), and (2) a restricted definition, which includes only physical and sexual assaults consistent with the BJS classification of IPV. All inferential analyses in this study, including logistic regression models, use the restricted definition to ensure conceptual and statistical alignment with previous work [24,27].

In the NCVS, only victims identified as having socio-emotional problems were asked about specific symptoms “for a month or more” following the crime. Emotional symptoms included feeling worried or anxious, angry, sad or depressed, vulnerable, violated, mistrustful, or unsafe. Physical symptoms included headaches, trouble sleeping, changes in eating or drinking, upset stomach, fatigue, high blood pressure, muscle tension, or back pain. We constructed two binary outcome variables from these fourteen items, following previous research [28]. For the first dependent variable, we coded cases as one if victims reported at least one emotional symptom and zero otherwise. For the second dependent variable, we coded cases as one if victims reported at least one physical symptom and zero otherwise. Victims who were screened as not having socio-emotional problems, that is, they reported no or only mild distress and no problems at work, school, or with family and friends, were not asked the follow-up symptom items; for these cases, both outcomes were coded as 0. Following BJS practice, we also imputed outcomes as 0 for respondents who skipped the socio-emotional screening and therefore were not asked about emotional or physical symptoms (n = 2545).

The primary predictors measure injury severity during the IPV incident. We coded any injury as one if either of two injury items indicated injuries (codes 2–11) and zero otherwise. We also coded whether the victim was knocked unconscious during the incident. Because the NCVS does not include neuroimaging or clinical assessments, loss of consciousness serves as the closest available proxy for probable traumatic brain injury (pTBI), a clinical marker of head trauma with potential neurological consequences [16]. We controlled a range of sociodemographic and contextual characteristics. Victim demographics included sex (female vs. male), age (modeled both continuously and categorically: <19, 19–39, 40–59, 60+), race/ethnicity (White [reference], Black, Hispanic, American Indian/Alaska Native, Asian/Pacific Islander, and multiracial), and education (high school or less vs. more than high school). Additional covariates included U.S. region (Northeast [reference], Midwest, South, West, and missing), population size of residence (<50,000 [reference], 50,000–250,000, ≥250,000), police reporting of the incident (yes/no), and survey year.

### Analytic Strategy

We implemented all descriptive and model-based estimates using survey methods. Specifically, we applied NCVS victimization weights in survey designs with independent primary sampling units to generate nationally representative estimates. First, we summarized yearly totals for all victimizations, violent victimizations, and IPV victimizations. We also looked at the share of violent crimes caused by IPV and the proportion of IPV incidents that resulted in injury. Next, we used survey-weighted bivariate logistic regressions to estimate the odds ratios (ORs) for reporting one or more emotional or physical symptoms.

We fit multivariable survey-weighted logistic regressions separately for emotional and physical symptoms. We modeled age and year with restricted cubic splines to allow flexible nonlinear associations (age knots at 25, 40, and 65; year knots at 1999, 2007, and 2015) following [26]. Models included all covariates listed above, along with injury status and loss of consciousness. From the multivariable models, we generated predicted probabilities on the response scale under prespecified scenarios. We averaged predictions across year, region, and population size where indicated, and reported results by sex, injury status, and race/ethnicity. For an injured-only subset, we contrasted those who reported losing consciousness after the IPV victimization with those who were not by sex or race. We also visualized marginal effects for age and year with 95% confidence intervals. We expect that IPV incidents involving injury will be associated with higher odds of reporting at least one emotional symptom and at least one physical symptom, and that, among injured survivors, loss of consciousness will be associated with greater risk. Further, we anticipated that females and persons from historically minoritized groups would have more physical and emotional symptoms compared to males and White survivors, respectively. Regarding age, we expect to see a nonlinear pattern, with symptom risk peaking in early to mid-adulthood.

We conducted all analyses in R (version 4.3.3) [29] using the survey [30], rms [31], dplyr [32], and ggplot2 [33] packages.

## 3. Results

The NCVS public-use data from 2009 to 2023 revealed that violent victimizations ranged from 1.7 to 2.5 million annually. Depending on the operational definition, IPV accounted for between 14% and 22% of violent crimes per year under the broad definition (all violent partner victimizations) and 19% to 31% under the restricted definition (partner assaults and sexual violence only). The restricted definition yields a higher national prevalence of IPV because it excludes partner-perpetrated robberies and threats, retaining only assaultive and sexual incidents. Removing these cases from both the numerator (IPV incidents) and denominator (all violent crimes) reduces the overall number of violent crimes more than the number of IPV assaults, thereby increasing the proportion of IPV among all violent victimizations. Across the study period, between 2% and 7% of IPV incidents meeting the restricted definition involved a coded injury, corresponding to an estimated 7000 to 39,000 injury-related IPV incidents nationally each year (weighted estimate). It is important to note that NCVS data collection was severely limited during the COVID-19 pandemic, affecting the interpretation of 2020–2021 estimates.

Table 1 shows the percentage of IPV survivors reporting at least one symptom overall and by sociodemographic characteristics. Overall, 70.6% of IPV victims of assault and sexual violence reported at least one emotional symptom, and 53.6% reported at least one physical symptom lasting a month or longer. Symptom rates varied widely based on victim characteristics. Among females, 76.1% reported emotional symptoms and 60.5% reported physical symptoms, compared to 52.7% and 31.0% among males. Younger victims indicated somewhat lower symptom prevalence than middle-aged or older victims; in contrast, American Indian/Alaska Native and Asian/Pacific Islander victims had the highest proportions of both emotional and physical symptoms. Injured victims had a notably higher prevalence: 81.4% reported emotional symptoms, and 72.6% reported physical symptoms.

The multivariable model results (Table 2) show that these differences remain after adjusting for covariates. Female sex was strongly linked to higher odds of symptoms (OR = 2.94 for emotional; OR = 3.61 for physical). Injury nearly doubled the odds of emotional symptoms (OR = 1.96) and tripled the odds of physical symptoms (OR = 2.79). Loss of consciousness was also significant, further increasing the likelihood of both outcomes. The effect of race/ethnicity was more variable. For example, Native American/Alaska Native victims had significantly higher odds of both outcomes, while Hispanic victims had lower odds of physical symptoms compared to non-Hispanic White victims. Education, region, and community size were not consistently related to symptoms. For clarity, the adjusted odds ratios from the multivariable logistic regression are visualized in Figure 1.

We used post-estimation predictions from multivariable models to examine how the probability of experiencing each type of symptom varied by survey year and victim age (Figure 2). Predicted probabilities of both physical and emotional symptoms increased during early and middle adulthood and then leveled off or declined slightly at older ages. Temporal patterns were modest, with symptom prevalence remaining relatively stable across survey years, exhibiting only minor upward fluctuations in more recent years.

Predicted probabilities further highlight the extent of sociodemographic and injury-related differences in symptoms following IPV assault (Figure 3 and Figure 4). Figure 3 presents model-based estimates for the full sample and shows consistently higher probabilities of both emotional and physical symptoms among survivors who sustained injuries, with higher risks for females across all racial and ethnic groups. For example, among White male survivors with no injury, the predicted probability of reporting physical symptoms was 0.26 (95% CI: 0.18–0.33), compared to 0.76 (95% CI: 0.65–0.87) among White female survivors who were injured. Emotional symptom probabilities exceeded 0.80 among injured female survivors across most racial and ethnic groups, highlighting the substantial burden of persistent psychological distress associated with injury.

Figure 4 isolates the subset of survivors who were injured and examines whether loss of consciousness, an indicator of pTBI, further heightens symptom risk. Across racial and ethnic groups, survivors who experienced loss of consciousness had markedly higher predicted probabilities of both emotional and physical symptoms compared to those who did not. These differences were most pronounced among women, more than 90% of whom were predicted to report emotional or physical symptoms following loss of consciousness. Complete model-based predicted probability estimates are presented in Appendix A.

## 4. Discussion

This study examined the short- to medium-term health consequences of IPV by linking incident-level victimization data to survivors’ reports of physical and emotional symptoms lasting at least one month. Using a nationally representative sample, we found that more than two-thirds of IPV survivors experienced at least one emotional symptom, and more than half experienced at least one physical symptom after victimization. The risks were substantially higher for survivors who sustained injuries, and especially for those who lost consciousness, with predicted probabilities of symptoms often exceeding 80 to 90 percent. These findings provide some of the most substantial evidence to date that IPV-related injuries, including pTBI, compound both physical and emotional burdens, and that these burdens vary by sociodemographic characteristics.

Our findings align with prior research showing that IPV is closely linked to physical health problems, emotional distress, and long-lasting trauma symptoms [1,13,34]. Hullenaar et al. examined outcomes across victim–offender relationships (family/intimate partner, acquaintance, stranger) and found that IPV victims faced greater risks of adverse outcomes, with injury further increasing the likelihood of reporting physical and emotional symptoms [35]. We extend this work by focusing exclusively on IPV and by examining how injury severity and loss of consciousness, which serves as a marker of pTBI, compound the risk of both emotional and physical symptoms. In addition, we estimate the national prevalence and predicted probabilities of any symptom across survey years, sex, race and ethnicity, and injury status, providing a more detailed and population-level view of the consequences of IPV on emotional and physical health.

Consistent with previous research [7,13], our results demonstrate that sustaining a serious injury, particularly one involving loss of consciousness, substantially increases the risk of enduring adverse emotional and physical outcomes that persist for months after the incident. Betrayal trauma helps explain why these symptoms persist, manifesting in fear and avoidance, altered relationship expectations, shame and low self-esteem, and communication difficulties. Research also demonstrates that betrayal trauma extends beyond interpersonal relationships to include institutions: when systems fail to provide safety and fair treatment, trauma symptoms worsen in measurable ways. Our results align with [36], who surveyed 123 clients of two agencies serving IPV survivors and found that fear of being blamed, feelings of shame, and lack of community support were significant barriers to seeking help. Silence rooted in shame can easily be misinterpreted as guilt or a lack of credibility, while hesitation to take protective actions may reflect fear, mistrust, or feelings of being unsafe rather than disengagement.

Further, we found that the prevalence of emotional and physical health symptoms has remained relatively stable over time, with only modest increases in recent years, suggesting little measurable progress in reducing the health consequences survivors experience after victimization. Age differences were noted for physical symptoms, which peak between ages 30 and 50 before declining modestly at older ages. In contrast, emotional symptoms remain consistently high across adulthood, with predicted probabilities above 70% even into older age. Consistent with prior research, the higher predicted probabilities observed for females, American Indian/Alaska Native, and Asian/Pacific Islander survivors are consistent with research documenting sex differences and racial/ethnic disparities in IPV exposure and general health outcomes [22], particularly surrounding experiences of shame [18,21]. Similarly, in the present study, the high prevalence of emotional symptoms among survivors of IPV, including worry, anxiety, anger, sadness, and feelings of vulnerability, violation, mistrust, and being unsafe, particularly among those who sustained injuries or pTBI, underscores the profound emotional toll of IPV beyond its physical consequences.

The increased likelihood of reporting emotional and physical symptoms may help explain why IPV survivors often experience disruptions in work, school, and family life [5,15]. Numerous studies show that exposure to interpersonal violence contributes to increased risks of depression, anxiety, and chronic health conditions later in life [1,37,38]. Our results add to this evidence by quantifying the additional risks associated with being injured or experiencing unconsciousness following an IPV-related physical or sexual assault, highlighting the importance of targeted screening and intervention for these high-risk groups.

### 4.1. Implications for Proactive Public Health Policy to Support IPV Survivors

The findings of this study underscore the need to integrate medical, neurological, and psychological screening into IPV response protocols, including within legal settings, especially for survivors who sustain injury, especially pTBI. Although federal initiatives encourage screening in healthcare settings, to date, no federal mandate requires IPV screening, and states maintain varied reporting laws, creating uneven requirements for providers [39,40], resulting in a lack of systematic screening [41]. Establishing a federal standard could help ensure consistency and equity in screening, which is critical, especially considering the variable definition of what constitutes IPV across states.

The study is interpreted through the lens of betrayal trauma, which helps explain the emotional consequences observed among IPV survivors. Betrayal trauma occurs when harm is inflicted by someone the survivor depends on for safety and trust, creating emotional distress characterized by fear, vulnerability, and mistrust. Although the NCVS does not measure trauma processes directly, the high prevalence of symptoms such as anxiety, sadness, mistrust, and feeling unsafe, particularly among survivors who sustained injuries or probable traumatic brain injury, suggests enduring emotional consequences consistent with betrayal-related responses. These findings indicate that IPV produces not only physical harm but also persistent emotional distress that reflects the violation of safety and trust central to intimate relationships.

Betrayal trauma includes emotional symptoms that IPV survivors experience, such as feeling worried or anxious, angry, vulnerable, violated, mistrustful, or unsafe. These symptoms carry not only medical consequences, including heightened risks of chronic stress and mental health disorders, but also broader social consequences for relationships, family stability, and institutional involvement in areas such as schooling, employment, and the legal system [16]. To address the persistent health problems revealed in our findings, particularly among survivors with injuries or loss of consciousness, and to reduce disparities by sex and race/ethnicity, legal and medical systems should adopt shame-informed approaches in their responses to IPV. One such model is Blueprint for Safety, a program that coordinates how agencies respond to domestic violence crimes [28]. The program enhances control over offenders, facilitates rapid intervention, connects survivors with resources, and shifts the burden from survivors to the system. Building on its demonstrated success in Saint Paul, Minnesota, and other U.S. cities, implementation at the federal level would standardize protocols across states, strengthen consistency in practice, and reduce inequities in access to protection and support.

Without awareness of shame’s emotional effects, institutions risk discrediting survivor claims or reinforcing harm through disbelief and procedural retraumatization—a process that constitutes institutional betrayal. Shame-informed approaches emphasize how shame functions as both an emotional and structural barrier to recovery. Shame can silence disclosure, reinforce self-blame, and deter help-seeking, particularly when survivors encounter judgmental or retraumatizing institutional responses. A shame-informed model, therefore, seeks to identify and reduce practices that elicit shame—such as blaming language, lack of transparency, or exposure to one’s abuser—and to promote responses that restore dignity, autonomy, and trust. For legal professionals and advocates, this involves recognizing behavioral and verbal expressions of shame, fostering trust through compassionate engagement, and centering the strengths of survivors rather than their perceived deficits [15]. Judges, in particular, hold significant power to shape survivors’ emotional experiences in the courtroom. Shame mitigation in judicial settings requires training judges to recognize how shame, trauma, and relational abuse intersect; to interpret emotional or avoidant behaviors as trauma responses rather than signs of deception or instability; and to cultivate dignity and safety for all litigants. Courts can reduce public exposure through measures such as limiting bulk hearings, allowing private testimony, or managing virtual hearings to prevent unwanted recording or audience humiliation [15].

Institutions must also be intentional about addressing what has been termed double betrayal—the compounding harm that occurs when the very systems designed to protect survivors perpetuate disbelief, exposure, or inequitable treatment [42]. Police reforms and courtroom procedures are especially critical, as law enforcement often serve as frontline responders and courts are the primary venues through which IPV cases are adjudicated [43]. Trauma-informed approaches such as closed courtrooms, remote testimony, or private communication with judges can reduce shame and promote survivor autonomy [15]. Attorneys and advocates likewise require training to recognize emotional responses, such as shame or guardedness, so that survivor protectiveness is not misinterpreted as apathy or dishonesty. Police interactions can intensify distress when survivors anticipate disbelief or blame, while courtrooms may become sites of retraumatization when survivors must share space with their abuser or respond to adversarial questioning. Building trust, using survivor-centered language, and safeguarding privacy are essential for validating survivors’ experiences and restoring agency. Judges should enforce zero-tolerance policies for disrespectful conduct and adapt courtroom layouts to minimize involuntary contact. For instance, providing private waiting areas, reducing reliance on mass hearings, and expanding the use of virtual platforms can reduce retraumatization. Abusers should never be allowed to confront or interrogate survivors directly, as such practices exacerbate trauma and undermine dignity. Collectively, these strategies transform institutional responses to IPV from systems that may replicate betrayal and shame to systems that validate survivors’ experiences, mitigate secondary trauma, and promote healing.

### 4.2. Limitations

Despite being the first study to examine the national prevalence of emotional and physical symptoms following IPV-related injury, this study is not without limitations. Because the NCVS relies on self-reported data, estimates may be influenced by recall bias, underreporting, or social desirability effects [44]. While our models document associations between IPV-related injury and subsequent symptoms, we cannot establish causal relationships, as preexisting vulnerabilities may contribute to symptom reporting. The time frame for symptom measurement, within six months of the crime, may underestimate the full scope of long-term or delayed effects. The survey also lacks detailed information on injury severity, types of head trauma, and important covariates such as prior mental health conditions, which limits interpretation. Data collection disruptions during the COVID-19 lockdown also affected estimates for 2020–2021. Finally, observed temporal changes could reflect broader social dynamics, such as shifting cultural attitudes and increased support for survivors, which may have encouraged greater disclosure.

## 5. Conclusions

Overall, this study contributes to the growing body of research showing that violence—especially IPV—is a significant risk factor for ongoing emotional and physical health issues, particularly among minoritized groups such as women. Understanding how injury severity and unconsciousness influence post-IPV health outcomes can help shape prevention strategies, early intervention efforts, and survivor-centered care models aimed at reducing the long-term impacts of IPV on individuals and communities.

## Figures and Tables

**Figure 1 ijerph-22-01829-f001:**
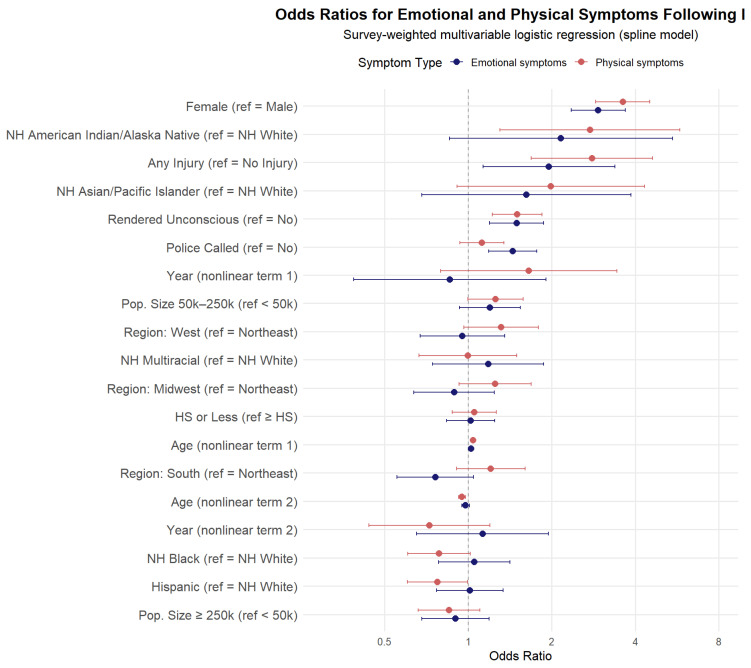
Adjusted Odds Ratios for Emotional and Physical Symptoms Following Intimate Partner Violence, 2009–2023. Survey-weighted multivariable logistic regression with nonlinear spline terms for age and year. Models adjust for sex, race/ethnicity, injury status, loss of consciousness, education, region, and population size. Reference categories are noted in parentheses. Error bars indicate 95% confidence intervals.

**Figure 2 ijerph-22-01829-f002:**
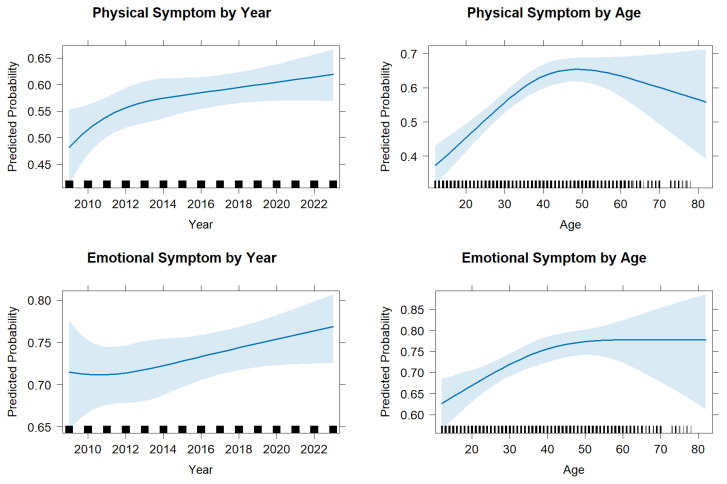
Temporal (**left**) and Age (**right**) Trends in Physical (**top**) and Emotional (**bottom**) Symptoms Following Intimate Partner Violence, 2009–2023. Note. Predicted probabilities of reporting at least one physical (**top row**) or emotional (**bottom row**) symptom among survivors of intimate partner violence in the United States, 2009–2023. Probabilities were estimated from survey-weighted logistic regression models and are shown as functions of survey year (**left panels**) and respondent age (**right panels**). Shaded bands represent 95% confidence intervals. Predictions were averaged across other covariates (sex, race/ethnicity, education, and region).

**Figure 3 ijerph-22-01829-f003:**
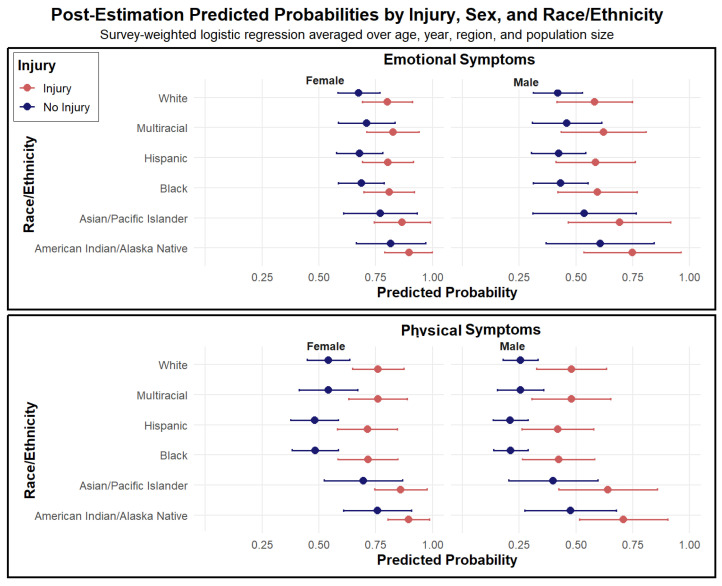
Post-Estimation Predicted Probabilities of Emotional and Physical Symptoms by Sex, Injury Status, and Race/Ethnicity. Note. Predicted probabilities and 95% confidence intervals from survey-weighted logistic regression models estimating the likelihood of reporting at least one emotional or physical symptom lasting one month or longer following intimate partner violence (IPV). Estimates are averaged over year, age, region, and population size. Separate models were fit for emotional and physical symptoms. Predicted probabilities are displayed by sex, race/ethnicity, and injury status (injury vs. no injury), with higher probabilities indicating greater prevalence of persistent post-victimization symptoms.

**Figure 4 ijerph-22-01829-f004:**
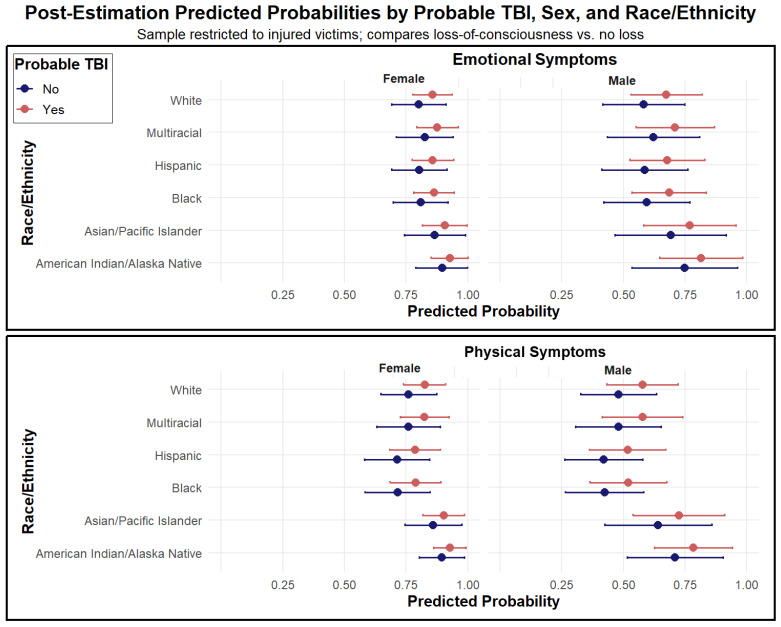
Post-Estimation Predicted Probabilities of Emotional and Physical Symptoms by Probable Traumatic Brain Injury (Loss of Consciousness), Sex, and Race/Ethnicity. Note. Predicted probabilities from survey-weighted logistic regression models showing the likelihood of reporting emotional (**top**) and physical (**bottom**) symptoms among injured IPV victims, by probable traumatic brain injury (pTBI; loss of consciousness), sex, and race/ethnicity. Estimates are averaged over age, year, region, and population size. Error bars indicate 95% confidence intervals. Probable TBI was defined as being “knocked unconscious or passing out” during the IPV incident.

**Table 1 ijerph-22-01829-t001:** Percentage of IPV Survivors Reporting at Least One Symptom by Sociodemographic Characteristics.

Characteristic	Level	Denom N (unwtd)	Denom N (wtd)	Emotional N (unwtd)	Emotional N (wtd)	Emotional % (wtd)	Physical N (unwtd)	Physical N (wtd)	Physical % (wtd)
Overall	All	3012	7,650,471	2186	5,403,579	70.6	1737	4,098,268	53.6
Age	Under 19	302	1,027,762	185	617,999	60.1	119	379,341	36.9
	19–39	1693	4,410,926	1227	3,141,339	71.2	976	2,380,037	54.0
	40–59	883	1,969,251	670	1,464,416	74.4	559	1,194,755	60.7
	60+	134	242,533	104	179,825	74.1	83	144,134	59.4
Race	Asian/PI	56	155,529	46	126,607	81.4	43	112,920	72.6
	Black	469	1,331,715	346	931,464	69.9	261	624,699	46.9
	Hispanic	492	1,304,346	357	933,534	71.6	270	648,830	49.7
	White	2290	5,609,273	1650	3,940,214	70.2	1314	3,048,489	54.3
	Multiracial	145	428,951	102	307,380	71.7	82	226,389	52.8
Sex	Male	668	1,800,398	372	949,016	52.7	237	558,707	31.0
	Female	2344	5,850,073	1814	4,454,563	76.1	1500	3,539,560	60.5
Highest Level of Education	HS diploma	1273	3,297,484	908	2,298,845	69.7	704	1,705,277	51.7
	>HS diploma	1720	4,311,041	1260	3,065,499	71.1	1019	2,366,506	54.9
Injury	No	2882	7,315,829	2076	5,126,300	70.1	1639	3,853,717	52.7
	Yes	130	334,642	110	277,279	82.9	98	244,551	73.1
Potential TBI	No	701	1,767,487	474	1,127,035	63.8	353	793,874	44.9
	Yes	2311	5,882,984	1712	4,276,544	72.7	1384	3,304,394	56.2
Region	Midwest	914	1,988,216	668	1,398,200	70.3	541	1,072,526	53.9
	South	1008	2,720,715	716	1,875,129	68.9	567	1,458,077	53.6
	West	720	1,841,674	528	1,317,647	71.5	425	1,005,487	54.6
	Northeast	370	1,099,866	274	812,603	73.9	204	562,178	51.1
Population Size	Under 50 K	1852	4,612,312	1340	3,235,453	70.1	1073	2,492,963	54.1
	50 K–250 K	631	1,630,186	477	1,193,815	73.2	387	931,333	57.1
	Over 250 K	529	1,407,973	369	974,312	69.2	277	673,971	47.9
Police Report	No	1336	3,271,726	932	2,191,525	67.0	768	1,709,629	52.3
	Yes	1626	4,258,017	1229	3,151,614	74.0	949	2,343,983	55.0

Note. Weighted estimates use the weight for victimization; percentages are survey-weighted. Key: N = number; “unwtd” = unweighted; “wtd” = weighted.

**Table 2 ijerph-22-01829-t002:** Emotional and Physical Symptoms After IPV Victimization (Multivariable ORs with Spline Terms).

	Reporting ≥ 1 Emotional Symptom			Reporting ≥ 1 Physical Symptom		
Predictor	Odds Ratio	CI Low	CI High	Odds Ratio	CI Low	CI High
—	—	—	—	—	—	—
Age (trend 1: 25–40)	1.024	1.010	1.038	1.043	1.029	1.056
Age (trend 2: 40–65)	0.979	0.949	1.010	0.951	0.925	0.977
Year (trend 1: 1999–2007)	0.859	0.387	1.906	1.653	0.795	3.437
Year (trend 2: 2007–2015)	1.127	0.653	1.944	0.725	0.439	1.197
Any Injury (Yes vs. No)	1.955	1.132	3.376	2.794	1.687	4.627
Female (Yes vs. No)	2.943	2.352	3.681	3.610	2.887	4.515
American Indian/Alaska Native (Yes vs. No)	2.162	0.857	5.458	2.748	1.303	5.798
Asian/Pacific Islander (Yes vs. No)	1.620	0.680	3.858	1.986	0.913	4.319
Black (Yes vs. No)	1.054	0.784	1.415	0.786	0.607	1.017
Hispanic (Yes vs. No)	1.014	0.769	1.337	0.776	0.605	0.996
Multiracial (Yes vs. No)	1.182	0.745	1.874	0.998	0.666	1.497
HS or Less (Yes vs. No)	1.022	0.836	1.249	1.054	0.878	1.265
Unconscious (Yes vs. No)	1.495	1.196	1.869	1.503	1.221	1.849
Population Size: 50 k–250 k	1.200	0.932	1.545	1.254	0.997	1.577
Population Size: ≥250 k	0.900	0.680	1.192	0.853	0.660	1.104
Police Called (Yes vs. No)	1.448	1.187	1.768	1.121	0.934	1.346
Region: Midwest	0.890	0.637	1.244	1.252	0.929	1.688
Region: South	0.762	0.555	1.046	1.208	0.908	1.608
Region: West	0.953	0.671	1.355	1.316	0.966	1.793

Note. Results are derived from survey-weighted logistic regression models predicting the likelihood of reporting at least one emotional or physical symptom following IPV. Year and age were modeled using restricted cubic splines with internal knots placed at 1999, 2007, and 2015 for year, and at ages 25, 40, and 65 for age. Reference categories include Region = Northeast and Population Size = <50,000. Binary predictors are coded as 1 versus 0. All estimates are weighted using the NCVS victimization weight to produce nationally representative results.

## Data Availability

The original data presented in the study are openly available at the Bureau of Justice Statistics, National Crime Victimization Survey website: https://bjs.ojp.gov/data-collection/ncvs#7-0 (accessed on 14 September 2025).

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
