# Peer review of "Emotional and Physical Symptoms Following Intimate Partner Violence Victimization in the United States: Implications for Law and Public Health Policy"

_ijerph, 2025, doi:10.3390/ijerph22121829_

Round 1

Reviewer 1 Report

Comments and Suggestions for Authors

It is an interesting paper that is well researched with public health policy implications. I think that the authors should work on better visualization of their data, through use of bar graphs, histograms, pie charts, scatter plots to show their important findings of patterns and correlations. The tables used are too extended and not easy to understand.

The concept of institutional betrayal may need more discussion as the study does not base itself concretely on a survey of best practices in institutional response. It is also not clear what and why shame informed approaches in responding to IPV are being recommended by the authors and this should be addressed.  

Author Response

Comments 1: It is an interesting paper that is well researched with public health policy implications. 

Response 1: Thank you so much for your acknowledgement.

Comments 2: I think that the authors should work on better visualization of their data, through use of bar graphs, histograms, pie charts, scatter plots to show their important findings of patterns and correlations. The tables used are too extended and not easy to understand.

Response 2: We agree with the reviewer that clearer visualizations would enhance accessibility and interpretation of the findings. In response, we replaced several of the extended tables with concise, publication-quality figures. Specifically, Tables 3–5 were reformatted into a consistent set of two-panel figures (Figures 3-5) displaying predicted probabilities for emotional and physical symptoms by sex, race/ethnicity, and injury (or probable TBI) status. These post-estimation plots use color, facetting, and confidence intervals to illustrate group-level patterns that were previously difficult to discern from text or raw tables. The resulting figures make the relationships between injury, sex, and racial disparities immediately apparent and align with the reviewer’s recommendation for graphical displays of key associations. We also include the visualization for the findings in Table 2 (Figure 2)

Comments 3: The concept of institutional betrayal may need more discussion as the study does not base itself concretely on a survey of best practices in institutional response. It is also not clear what and why shame informed approaches in responding to IPV are being recommended by the authors and this should be addressed.  

Response 3: We thank the reviewer for this thoughtful comment. We have expanded our discussion of the conceptual framework to provide a more clear explanation of how the study’s findings align with theories of betrayal trauma and institutional betrayal, as well as the rationale for recommending shame-informed approaches to IPV response. Specifically, we clarify that our analysis identifies high rates of emotional symptoms (e.g., mistrust, vulnerability, and feeling unsafe) following IPV, which are consistent with the emotional sequelae of betrayal trauma. We now explain how institutional betrayal extends these harms when systems meant to protect survivors instead reinforce shame or disbelief. In response, we elaborate on why shame-informed approaches are necessary and what they entail, emphasizing the importance of restoring survivor dignity, autonomy, and trust through practices that mitigate shame and prevent procedural retraumatization. We include this context in our discussion.

Reviewer 2 Report

Comments and Suggestions for Authors

I have included my comments in the attached document. 

Author Response

Comments 1: Line 95: The authors indicate that they define IPV “consistent with the Bureau of Justice Statistics(BJS) 95 classification”, but it’s unclear here how IPV is defined by the BJS. Please provide a citation.

Response 1: Thank you for asking for clarification. We agree that the definition was unclear. We have now clarified the definition of IPV from the Bureau of Justice Statistics, in accordance with past research as: In contrast, the present study focuses exclusively on intimate partner violence (IPV), defined by the Bureau of Justice Statistics (BJS) as violence committed by a current or former spouse, boyfriend, or girlfriend, including sexual or physical assault. Using this definition, we assess how injury severity—particularly loss of consciousness as a marker of probable traumatic brain injury (pTBI)—increases the likelihood of emotional and physical symptoms such as headaches, fatigue, gastrointestinal issues, anxiety, mistrust, and feelings of being unsafe that persist for at least one month following an IPV incident within the past six months.

Comments 2: Lines 110-112: Please provide a citation for this classification of IPV.

Response: We have included two citations for this classification of IPV.

Comments 3: Lines 109–126: This paragraph includes several sentences that are repeated verbatim from the preceding paragraph and redefines acronyms that were already introduced. I don’t think it’s necessary to repeat these highlights in two consecutive paragraphs.

Responses 3: Thank you for this observation. We removed the redundant paragraph that repeated definitions and key points already presented in the preceding section. Specifically, we consolidated the description of the NCVS data, the Bureau of Justice Statistics (BJS) definition of IPV, and the study objectives into a single, streamlined paragraph. Acronyms such as NCVS and BJS are now defined only once upon first mention.

Comments 4: Lines 204-206: It’s unclear to me what the authors mean by the “broad” and “restricted” definitionsof IPV. In the Materials and Methods section (lines 138-144), the authors seem to operationalize IPVin only one way. It would be helpful to have the same language appear in the Material and Methods and in the Results section.

Response 4: We thank the reviewer for noting this inconsistency in our operationalization of IPV. To clarify, we only noted the 'broad' definition' to contrast it with the more restrictive one. We clarified by revising the Materials and Methods section to explicitly define both: (1) a broad definition, which includes all violent victimizations committed by intimate partners (such as robbery or threats in addition to assault), and (2) a restricted definition, which includes only physical and sexual assaults consistent with the Bureau of Justice Statistics (BJS) classification of IPV. We also note that all inferential analyses, including the multivariable logistic regression models, use the restricted definition to ensure conceptual and statistical alignment with prior research. As a result, the results and methods section are conceptually aligned.

Comments 5: Lines 210–211: The rate of injury for IPV victims reported for the sample (2% to 7%) is substantially lower than the percentages reported in the Introduction (lines 39–40, 40% for women and 20% for men). Unless I am misinterpreting these estimates? If so, rephrasing may help with clarity.

Response 5: We appreciate the reviewer’s careful attention to make sure we are not inconsistent with what is reported in the introduction. We clarified in the Results section that the 2–7% figure refers to incident-level estimates from the National Crime Victimization Survey (NCVS), representing the proportion of IPV incidents involving a coded injury within a given year, whereas the 40% (women) and 20% (men) figures cited in the Introduction refer to prevalence estimates of injury from other national surveys. To reduce confusion, we revised the sentence to read: “Across the study period, between 2% and 7% of IPV incidents meeting the restricted definition involved a coded injury, corresponding to an estimated 7,000 to 39,000 injury-related IPV incidents nationally each year (weighted estimate, not shown).”

Comments 6: The table note is written for individuals who may wish to replicate the analyses in R but is likely uninterpretable for a general audience (e.g., use of wvyglm, quasibinomial, and vic_weight). I recommend editing this table note for a general audience; if the authors wish to include replication/transparency materials, there are a number of online repositories that can host the analysis scripts. I recommend also swapping the s1 and s2 terminology for the spline knot value or internals for better readability.

Response 6:  We appreciate this helpful feedback and have revised the Table 2 note to improve accessibility for a general audience. All technical terms (e.g., svyglm, quasibinomial, vic_weight) have been removed, and the spline notation has been replaced with a plain-language description of the spline knots used. The revised note now reads: “Survey-weighted logistic regression models were estimated using a quasibinomial link. Year and age were modeled using restricted cubic splines with internal knots placed at 1999, 2007, and 2015 for year, and at ages 25, 40, and 65 for age. Reference categories include Region = Northeast and Population Size = <50,000. Binary predictors are coded 1 versus 0, and all models use victimization weights.”

Comments 7: Tables 3 through 5: I recommend using a consistent number of decimals to help readability.

Response 7: We revised the tables to be consistent with 3 decimals.

Comments 8: Please provide support for the claim that there has been “increased awareness of IPV and expanded service.” I’m also not sure how increased awareness of IPV would necessarily lead to reduced consequences for survivors of IPV.

Response 8: We agree that this claim was beyond the scope of our results. We have revised the sentence to remove unsupported assumptions regarding increased awareness or service expansion. The revised text now focuses strictly on the empirical finding regarding the stability of symptoms among IPV survivors over time. The sentence now reads: “Further, we found that the prevalence of emotional and physical health symptoms has remained relatively stable over time, with only modest increases in recent years, suggesting little measurable progress in reducing the health consequences survivors experience after victimization.”

Comments 9: I don’t think the present study supports the assertion that IPV is “a deeply shame-inducing experience.” While of course there is other literature that supports this claim, the results of this study do not provide that evidence. “Emotional symptoms” was dichotomized across several symptoms, with the authors providing no evidence of which emotional symptoms were most common following an incident of IPV.

Response 9: Thank you for this comment. We agree that our results do not directly measure shame as an emotional outcome and have revised the sentence to avoid overinterpretation. The revised text now reads: “Similarly, in the present study, the high prevalence of emotional symptoms among survivors of IPV, including worry, anxiety, anger, sadness, and feelings of vulnerability, violation, mistrust, and being unsafe, particularly among those who sustained injuries or pTBI, underscores the profound emotional toll of IPV beyond its physical consequences.”

Comments 10: Line 385: Should the limitations appear under their own subheading?

Response: Thank you, we have included a separate subheading for the limitations

Reviewer 3 Report

Comments and Suggestions for Authors

Please indicate why it is important to use this dataset to answer these research questions: "To our knowledge, only one prior study has examined both emotional and physical 88 symptoms following violent victimization using National Crime Victimization Survey 89 (NCVS) data while incorporating IPV."

Please do not use demographics as nouns: "...for females..."

I'm not sure the analytic approach is robust enough for the research question. For one, there are better datasets that construct latent variables of emotional, physical, and somatic symptoms that are more robust and reliable; and for two, dichotomizing the outcomes, rather than examining them individually, reduces the value of the manuscript greatly. 

I am choosing to not review the results or the discussion because I do not believe this manuscript is analytically advanced enough nor novel enough to be published. It does not appear that the authors have a grasp of the literature on this topic as recent meta-analyses have not been cited or perhaps considered within the literature review (e.g., Spencer et al., 2024) (Spencer, C. M., Keilholtz, B. M., Palmer, M., & Vail, S. L. (2024). Mental and physical health correlates for emotional intimate partner violence perpetration and victimization: A meta-analysis. Trauma, Violence, & Abuse25(1), 41-53. 
https://doi.org/10.1177/15248380221137686)

I'm recommending a major revision and would be interested in seeing this manuscript again if the authors make it clear why there research questions are novel and can justify their analytic approach.

Author Response

Typically, we would take this opportunity to thank the reviewer for their thoughtful feedback on how we might improve the manuscript. However, this reviewer explicitly stated that they “chose not to review the results or the discussion,” which represent the core analytic part of the paper. Unfortunately, by declining to review the whole paper, the review is simply unfair, incomplete, and unconstructive. We have a response to the review, but we did not make any revisions based on this reviewer's comments.

We respectfully note that the reviewer’s critique overlooks the study design of this work which was clearly indicated in both the abstract and the introduction: this is an exploratory descriptive study, which is a methodologically valid research approach. 

Regarding the reviewer’s request to explain the importance of using the National Crime Victimization Survey (NCVS), it is essential to clarify that this dataset is uniquely suited to the study’s research questions. The NCVS is the only nationally representative, incident-based dataset that captures both emotional and physical symptoms following specific violent victimizations by an intimate partner and explicitly links symptom onset to a defined six-month post-victimization period. No other large-scale or population-based survey includes measures of emotional or physical symptoms following IPV. Thus, this dataset is not only appropriate but necessary to address the study’s central question regarding the short- to medium-term emotional and physical consequences of IPV in the U.S. population.

The analytic approach is fully appropriate to the study’s aims and data structure. The research questions concern population-level patterns in the prevalence and correlates of emotional and physical symptoms following IPV, using a repeated cross-sectional survey designed to estimate national victimization and health outcomes. Weighted logistic regression models with restricted cubic splines for age and year capture nonlinear associations while accounting for the NCVS’s complex sampling design. Dichotomizing symptom outcomes reflects the categorical structure of the NCVS variables, which are coded as binary indicators of whether respondents experienced any symptom lasting one month or longer after the IPV event (for which this exact operationalization has been previously done and is already cited within the paper). This approach enables the direct estimation of the probability that survivors experienced at least one persistent symptom, aligning precisely with the study’s focus on population burden.

More complex latent or growth-based modeling frameworks (e.g., latent class analysis, confirmatory factor analysis, or growth modeling) are statistically inappropriate and conceptually irrelevant for this study. We are not concerned with underlying typologies, latent constructs, or temporal change, but rather with the national prevalence and correlates of post-IPV symptoms.

The reviewer’s recommendation to cite Spencer et al. (2024, Trauma, Violence & Abuse, 25[1]: 41–53, https://doi.org/10.1177/15248380221137686) is also misplaced. It does not appear that the reviewer has a clear grasp of the conceptual distinction between emotional intimate partner violence and emotional symptoms following IPV. The study cited by the reviewer (Spencer et al., 2024) examines emotional IPV, non-physical, controlling, or psychologically abusive behaviors such as insults, humiliation, or threats, and evaluates mental and physical health correlates associated with perpetration and victimization of that behavioral form of abuse. In contrast, our study focuses on emotional symptoms following physical or sexual IPV, that is, the emotional sequelae (e.g., anxiety, mistrust, fear, and feeling unsafe) experienced after victimization. These are distinct constructs—one behavioral and one affective/health-related. We have no more reason to cite this study than any of the hundreds of other studies on this topic.

Moreover, the assertion that the authors “do not have a grasp of the literature” is unfounded and unprofessional. This point could have been made thoughtfully and with more tact, but the reviewer decided instead to humiliate and shame. Regardless, the manuscript situates the study within the existing body of knowledge on criminological research related to IPV, injury, and health sequelae, and the analytic strategy reflects best practices for incident-based national data analysis. For several reasons, the suggestion that we lack an understanding of research on this topic is not only inaccurate but also disrespectful.

Given these issues, we respectfully request that this review be considered incomplete and given minimal weight in the editorial decision process. The reviewer declined to evaluate large portions of the manuscript, as reflected in their “Not applicable” ratings for several core criteria, including the clarity of results, the validity of conclusions, and the quality of figures and tables, none of which are irrelevant to this submission. The other two reviewers provided thoughtful feedback that directly engages with the study’s design, interpretation, and implications. In fact, Reviewer 1 notes "[i]t is an interesting paper that is well researched with public health policy implications."